# Paternal Preconception Every-Other-Day Ethanol Drinking Alters Behavior and Ethanol Consumption in Offspring

**DOI:** 10.3390/brainsci9030056

**Published:** 2019-03-06

**Authors:** Erik Beeler, Zachary L. Nobile, Gregg E. Homanics

**Affiliations:** 1Department of Anesthesiology and Perioperative Medicine, University of Pittsburgh School of Medicine, 6068 Biomedical Science Tower-3, 3501 Fifth Avenue, Pittsburgh, PA 15261, USA; esb37@pitt.edu; 2Department of Biological Sciences, University of Pittsburgh, 4249 Fifth Avenue, Pittsburgh, PA 15260, USA; zachnobile@yahoo.com or zln5@pitt.edu; 3Center for Neuroscience, University of Pittsburgh School of Medicine, 6060 Biomedical Science Tower-3, 3501 Fifth Avenue, Pittsburgh, PA 15261, USA; 4Department of Pharmacology & Chemical Biology, University of Pittsburgh School of Medicine, 6060 Biomedical Science Tower-3, 3501 Fifth Avenue, Pittsburgh, PA 15261, USA; 5Department of Neurobiology, University of Pittsburgh School of Medicine, 6060 Biomedical Science Tower-3, 3501 Fifth Avenue, Pittsburgh, PA 15261, USA

**Keywords:** alcohol, EtOH, alcohol use disorder, alcoholism, epigenetics, epigenetic inheritance, intergenerational effect

## Abstract

Alcohol use disorder is a devastating disease with a complex etiology. Recent preclinical studies have revealed that paternal preconception chronic intermittent ethanol (EtOH) exposure via vaporized EtOH altered drinking behaviors and sensitivity to EtOH selectively in male offspring. In the current study, we used a voluntary oral route of paternal preconception EtOH exposure, i.e., intermittent every-other-day two-bottle choice drinking, and tested offspring for behavioral alterations. Fifteen EtOH drinking sires and 10 control sires were mated to EtOH naïve females to produce EtOH-sired and control-sired offspring. These offspring were tested using the elevated plus maze, open field, drinking in the dark, and unlimited access two-bottle choice assays. We found that paternal preconception every-other-day two-bottle choice drinking resulted in reduced EtOH consumption selectively in male offspring in the drinking in the dark assay compared to control-sired offspring. No differences were detected in either sex in the unlimited access two-bottle choice and elevated plus maze assays. Open field analysis revealed complex changes in basal behavior and EtOH-induced behaviors that were sex specific. We concluded that paternal preconception voluntary EtOH consumption has persistent effects that impact the next generation. This study adds to a growing appreciation that one’s behavioral response to EtOH and EtOH drinking behavior are impacted by EtOH exposure of the prior generation.

## 1. Introduction

Alcohol use disorder (AUD) is a prevalent health issue with a major impact on numerous aspects of society. This disorder is defined by the National Institute on Alcohol Abuse and Alcoholism as “a chronic relapsing brain disease characterized by an impaired ability to stop or control alcohol use despite adverse social, occupational, or health consequences.” AUD is estimated to affect 15.1 million adults aged 18 years and older and 623,000 adolescents aged 12–17, with an annual economic burden of $249 billion in the US alone [1,2]. Previous research has shown that the prevalence of this disease is increasing, with a preponderance in the 18–30 year old age group [3]. This disease is further complicated by the heterogeneity of phenotypes, each of which may have a unique biological and environmental explanation [4,5]. As such, it’s important that we better our understanding of the pathogenesis of this disease to better treat it.

AUD has consistently been shown to have a high heritability of ~50% [4,5,6]. Genome wide association studies have been widely utilized to search for the genetic source of this significant heritability (e.g., 5–10). Although these efforts have resulted in the identification of a few relevant single nucleotide polymorphisms, these variants only explain a very small fraction of the heritability shown in twin and adoption studies [7,8,9,10,11,12]. Although there are several plausible explanations for the great difficulty that has been encountered in identifying the DNA variants responsible for AUD heritability, some recent research has turned to other forms of germline inheritance that do not involve changes in the DNA sequence, namely epigenetic inheritance. 

Epigenetic inheritance is the transfer of acquired traits across generations via germline transmission of epigenetic variants and has received attention as it has been shown to be involved in a number of behavioral and physiological phenotypes in offspring [13,14,15,16]. Although several examples of epigenetic inheritance in humans have been reported [17,18,19,20], such human studies are difficult to conduct and interpret. In contrast, laboratory rodents are apt for epigenetic study as they afford strict environmental control and inbred lines allow for the near elimination of genetic variation. Notable examples of epigenetic inheritance observed in rodents includes inheritance of obesity and reproductive disease following parental hydrocarbon exposure [21], conditioned responses to odors [15], stress response [22], depressive-like and risk-taking behaviors [23], insulin resistance [24], and drug use behaviors [14].

Similar studies in rodents have indicated that paternal preconception alcohol can also have effects that impact multiple behavioral and physiological phenotypes in subsequent generations, presumably via epigenetic inheritance (for reviews, see [5,25]). Observed examples of these intergenerational effects of EtOH include expression of an ADHD-like phenotype and dopamine transporter dysfunction [26], changes in development time [27], increased anxiety-like and depressive-like behaviors as well as cognitive impairment [28], and decreased testosterone [29]. Of direct relevance to the work reported here, we previously demonstrated that paternal preconception exposure of C57BL/6J (B6) sires to chronic intermittent EtOH via EtOH vapor inhalation followed by mating to Strain 129 Sv/ImJ [30] or B6 [31] EtOH naïve females resulted in sex specific phenotypes in offspring. Specifically, EtOH-sired (E-sired) male offspring had reduced EtOH preference and consumption in a home cage unlimited access two bottle choice (2BC) assay compared to control-sired (C-sired) males. E-sired males also showed increased sensitivity to EtOH’s anxiolytic and motor enhancing effects when performing elevated plus maze (EPM) and rotarod behavioral assays, respectively, compared to C-sired offspring [30]. Importantly, Rompala, et al. [32] found that paternal preconception chronic variable stress resulted in similar behavioral changes in offspring, with a reduction in EtOH preference and consumption in stress-sired males compared to C-sired males.

Because CIE EtOH vapor exposure can be considered stressful, it is unclear if the altered phenotypes observed in Finegersh and Homanics [30] and Rompala et al., [31] are due to a direct effect of EtOH or due to an effect of stress. Furthermore, it is unknown how closely studies that utilize forced vapor exposure relate to human alcohol exposure which is via a voluntary oral route. To investigate these issues, we tested the hypothesis that paternal preconception EtOH exposure via voluntary oral intake would induce a similar phenotype in offspring as has been observed following vapor exposure.

## 2. Materials and Methods

### 2.1. Animals

All experiments were approved by the Institutional Animal Care and Use Committee of the University of Pittsburgh.

Specific pathogen free male (*n* = 31) and female (*n* = 30) B6 mice (The Jackson Laboratory; Bar Harbor, ME, USA) were housed in a temperature-controlled AAALAC-approved rodent facility in cages with nestlets (Ancare, Bellmore, NY, USA) and Innodomes (Innovive, San Diego, CA, USA) under reverse light cycle (lights out at 10 am and on at 10 pm). An overview of the experimental design and the procedures the mice were exposed to is shown in Figure 1.

### 2.2. Every-Other-Day 2 Bottle Choice (EOD 2BC) EtOH Exposure

The EOD 2BC assay was chosen because it is a widely used method that induces a rapid escalation in voluntary EtOH consumption in B6 mice [33,34,35]. Thirty-one, 3-week old EtOH naïve B6 male mice were singly housed and allowed to acclimate to drinking from two sipper tubes containing only water for a week. At four weeks of age, 21 of these mice were introduced to EOD 2BC EtOH drinking by offering 3%, 6%, and 10% EtOH (Decon Labs, Inc., King of Prussia, PA, USA) for a 24 h period in one of the 2 drinking tubes on Monday, Wednesday, and Friday, respectively during the first week. During all 11 subsequent weeks, 20% EtOH was provided for a 24 h period on Monday, Wednesday, and Friday in one of the drinking tubes. Bottle sides were switched during each EtOH exposure to avoid side preference. Amount consumed from each drinking tube was measured at the end of each drinking period (~10–11:30 a.m.) with values being used to calculate total volume of fluid ingested (mL), consumption of EtOH (g/kg/day), and preference for EtOH (EtOH intake/total volume fluid consumed). Control sires consisted of 10 EtOH naïve B6 male mice that were housed and manipulated under identical conditions to the EtOH exposed sires except that both drinking tubes always contained only water. Body weights were recorded every Friday. 

### 2.3. Selection of Sires for Breeding

Before breeding, male mice underwent a 48 h washout period to allow EtOH to be cleared. Fifteen of the 21 EtOH consuming sires were selected for breeding based on their consistently high levels of EtOH consumption and preference. Each male was caged with 2 EtOH naïve 8-week-old B6 females for 72 h. The 10 control mice underwent the same breeding protocol. After the 72 h period, sires were sacrificed, and 2 dams remained housed together for 2 weeks at which point each female was subsequently individually housed. Litters were weaned at 3 weeks. A total of 172 offspring were weaned, and starting at 6 weeks of age 120 of these offspring were singly housed in cages with two sipper tubes of water in order to acclimate the mice to drinking from sipper tubes. 

### 2.4. Behavioral Assays

The 120 offspring were evenly split into two cohorts of 60, with each cohort consisting of 15 C-sired males, 15 C-sired females, 15 E-sired males, and 15 E-sired females. These animals were pseudo randomly chosen from each litter with no more than 2 of each sex being used from each litter to prevent litter effects from acting as a confounding variable. All 120 of these mice underwent both EPM and open field (OF) testing at 8 weeks of age, with cohort 1 receiving an EtOH injection (0.75 g/kg for males and 1 g/kg for females) and cohort 2 receiving saline (0.9%; 0.02 mL/g body weight) injections intraperitoneally. EtOH injection dosages were determined utilizing EPM pilot study results (not shown). 

Lighting was kept low in the behavioral room. The habituation and behavioral assays were performed during the last 5 h of their light cycle (~5–10 a.m.) Mice were moved into the testing room with food and water an hour before the assays to acclimate. They were then injected and given a ten-minute waiting interval in their home cage before engaging in an OF 5 min session and then moved immediately to EPM for another 5 min session. 70% EtOH was used to clean assay apparatuses between subjects. 

The OF assay was used to analyze motor and anxiety-like behaviors. Mice were placed into the center of a 43.2 × 43.2 × 30.5 cm OF box with a white floor and clear plexiglass walls (Med Associates Inc., St. Albans, VT, USA) at the beginning of the assay. OF testing took place in sound attenuation boxes (Med Associates Inc.) lit by a 1-watt bulb to prevent lighting differences and using a small fan as background noise to prevent sounds from affecting results. Scoring occurred via infrared beam breaking utilizing Activity Monitor version 5 (Med Associates Inc.) with the results being broken down into center vs periphery activity. The center of the box was defined as a centered square of 11.25 cm × 11.25 cm. The periphery consisted of all areas outside of that zone. 

The EPM was utilized to analyze anxiety-like behaviors. Mice were placed in the center of the EPM facing the open arms of the apparatus 15 min after initial injection. The assay was recorded with a video camera for 5 min and subsequently scored manually. Number of arm entries and time spent in each arm were recorded, with arm entries being counted only if all four limbs were in the arm. Those mice that fell off the open arms were eliminated from the analysis. Due to technical issues with video recording, some mice were unable to be scored on EPM.

### 2.5. Drinking in the Dark Assay

One week after OF and EPM testing, Cohort 1 underwent the drinking in the dark (DID) assay [36]. Mice were offered 20% EtOH from one sipper tube for 2 h each day for three consecutive days followed by a 4 h period on the fourth day. Each test period started 3 h into the dark cycle. Measurements were taken at the end of each drinking session with values being used to calculate EtOH consumption (g/kg/test period). At the end of the 4 h session, blood samples were taken from a tail nick to determine blood EtOH concentrations (BECs) using an Analox Alcohol Analyzer (Analox Instruments, Lunenberg, MA, USA).

### 2.6. Continuous 2BC Drinking Assay

One week following OF and EPM testing, Cohort 2 was tested for continuous 2BC EtOH drinking by replacing one of the water tubes with one containing 3% EtOH. EtOH concentration was increased 3% every 4 days (3–21% tested). Cages were changed, and body weights taken on the day EtOH concentration changed. Subjects had ad libitum access to food, water, and EtOH. Every 48 h, amount consumed from each drinking tube was recorded and used to calculate total volume of fluid consumed (mL), EtOH consumption (g/kg/day), and EtOH preference (EtOH intake/total volume). Tube position was switched every 48 h to prevent side preferences.

Preference and consumption of a sweet (0.06% saccharin) and bitter (0.6 mM quinine) solution was measured for two days each with a one-week washout period between tastants in a similar 2BC drinking assay. 

### 2.7. Statistical Analysis

Male and female mice were analyzed separately because of well-known differences in various behavioral responses to EtOH and in EtOH drinking behaviors. Data were analyzed with two-way ANOVAs in Graphpad Prism except as follows. Unpaired T tests were used for DID. Repeated measures ANOVA was used for body weight and 2BC analysis with time and concentration as the within-subject factors, respectively. Secondary analysis consisted of pre-planned comparisons of significant results utilizing Sidak tests. *p* < 0.05 was considered statistically significant. 

## 3. Results

### 3.1. Paternal Preconception EOD 2BC Drinking

EOD 2BC access to EtOH in the F0 generation was maintained for approximately three months (Figure 1). Based on visual inspection of drinking data from individual mice, 15 males were selected for breeding which showed consistently high levels of EtOH consumption (Figure 2A,B,E) and preference (Figure 2C,D). Six sires were not used for breeding. All E-sires experienced a quick increase and ultimate plateau in consumption and preference consistent with prior studies [15,16]. Average daily EtOH consumption for each E-sire across the entire exposure period is shown in Figure 2E. Total fluid intake (Figure 2F) was not different between control or EtOH drinking sires. 

Although there was a significant effect of time on sire body weight (F_(10,280)_ = 282.9, *p* < 0.0001), there was no effect of EtOH or EtOH by time interaction (data not shown). 

### 3.2. Breeding Results

Eight of 10 (80%) control sires produced litters, with four of the eight males siring two litters. Similarly, 11 of 15 (73%) EtOH drinking sires produced litters, but only three of the 11 males sired two litters. In total, 81 (43 males/38 females) C-sired and 91 (49 males/42 females) E-sired offspring were produced that survived to weaning age. There was no significant effect of EtOH exposure on litter size, number of litters, or sex ratio (data not shown). From these offspring, 120 were selected for behavioral studies as detailed above.

### 3.3. Offspring Behavioral Assays

#### 3.3.1. Open Field

For F1 male mice in the OF assay, there was no effect of sire, treatment, or sire X treatment interaction for total ambulatory distance (Figure 3A), ambulatory distance in the periphery (Figure 3B) or ambulatory time in the periphery (Figure 3C). In contrast, there was a significant effect of treatment (F_(1,56)_ = 11.77, *p* < 0.01), but not sire or sire X treatment interaction on ambulatory distance in the center of the OF (Figure 3D). Pre-planned comparisons revealed that EtOH injection reduced ambulatory distance in the center in the E-sired male offspring (*p* < 0.01). Similarly, treatment also had a significant effect (F_(1,56)_ = 7.424, *p* < 0.001) on ambulatory time in the center of the OF (Figure 3E); there was no effect of sire or sire X treatment interaction. Pre-planned comparisons revealed that EtOH injection of E-sired males reduced ambulatory time in the center compared to saline injected E-sired males (*p* < 0.01). For the percentage of time spent in the center of the OF (Figure 3F), there was a significant effect of treatment (F_(1,56)_ = 35.36, *p* < 0.0001), sire (F_(1,56)_ = 39.56, *p* < 0.0001) and sire X treatment interaction (F_(1,56)_ = 15.09, *p* < 0.001). Pre-planned comparisons revealed that saline injected E-sired males spent a significantly larger percentage of their time in the center compared to saline injected C-sired males (*p* < 0.0001) and EtOH injected E-sired males (*p* < 0.0001).

For F1 females in the OF, there was an effect of treatment (F_(1,56)_ = 7.895, *p* < 0.01) but not sire or interaction of treatment X sire on total ambulatory distance (Figure 4A). Pre-planned comparisons revealed that compared to saline injection, EtOH injection increased total ambulatory distance in C-sired offspring (*p* < 0.05). Ambulatory distance in the periphery (Figure 4B) and ambulatory time (Figure 4C) in the periphery of the OF similarly showed an effect of treatment (F_(1,56)_ = 8.73, *p* < 0.01 and F_(1,56)_ = 14.63, *p* < 0.001, respectively), but no sire or sire X treatment interactions. Further analysis revealed that compared to saline injected C-sired females, EtOH injected C-sired females showed a significant increase in ambulatory distance and time in the periphery (*p* < 0.05 and *p* < 0.01, respectively). Ambulatory distance in the center (Figure 4D), ambulatory time in center (Figure 4E), and percentage of time in center (Figure 4F) of the OF were not affected by treatment, sire, or sire X treatment interaction.

No significant differences in male total resting time in the OF were found (Figure 5A). However, both resting time in the periphery and center of the OF showed significant differences. Resting time in the periphery (Figure 5B) showed significant effects of treatment (F_(1,56)_ = 20.32, *p* < 0.0001), sire (F_(1,56)_ = 10.18, *p* < 0.01), and interaction between the two variables (F_(1,56)_ = 5.289, *p* < 0.05). Pre-planned comparisons showed that saline injected E-sired males rested significantly less in the OF periphery than saline injected C-sired males (*p* < 0.001) and EtOH injected E-sired males (*p* < 0.0001). This is complemented by significant effects of sire (F_(1,56)_ = 27.93, *p* < 0.0001), treatment (F_(1,56)_ = 42.45, *p* < 0.0001), and interaction between the two variables (F_(1,56)_ = 18.81, *p* < 0.0001) for resting time in the center (Figure 5C). Pre-planned comparisons identified significantly greater resting time in saline injected E-sired males when compared to saline injected C-sired males (*p* < 0.0001) and EtOH injected E-sired males (*p* < 0.0001). The percentage of time in the center resting (Figure 5D) was significantly affected by sire (F_(1,56)_ = 19.97, *p* < 0.0001) and treatment (F_(1,56)_ = 6.322, *p* < 0.05), but there was no interaction between the two. Pre-planned comparisons revealed that saline injected E-sired males were found to spend a larger percentage of their time in the OF center resting when compared to saline injected C-sired males (*p* < 0.0001) and EtOH injected E-sired males (*p* < 0.01).

Female total resting time (Figure 6A) was significantly affected by treatment (F_(1,56)_ = 7.695, *p* < 0.01), but not sire or interaction. Pre-planned comparisons revealed a small but significant reduction in total resting time in EtOH injected C-sired females compared to saline injected C-sired females (*p* < 0.05). Resting time in the periphery of the OF (Figure 6B) was affected by treatment (F_(1,56)_ = 4.576, *p* < 0.05), but not sire or interaction between the two. Pre-planned comparisons revealed no significant differences. Time resting in center (Figure 6C) and percentage of time resting in center (Figure 6D) of the OF were not impacted by treatment, sire, or sire X treatment interaction.

For male offspring, total vertical time in the OF (Figure 7A) was significantly affected by sire (F_(1,56)_ = 4.168, *p* < 0.05) and treatment (F_(1,56)_ = 4.529, *p* < 0.05) but not by the interaction between the two. Pre-planned comparisons did not reveal any significant relationships. The time spent vertically in the periphery of the OF (Figure 7B) was significantly affected by sire (F_(1,56)_ = 11.75, *p* < 0.01), but not by treatment or interaction of sire X treatment. Further analysis revealed saline injected E-sired males spent significantly less time (*p* < 0.05) in the periphery vertical compared to saline injected C-sired males. Time spent vertical in the center of the OF (Figure 7C) was significantly affected by sire (F_(1,56)_ = 47.52, *p* < 0.0001), treatment (F_(1,56)_ = 47.52, *p* < 0.0001), and interaction between the two (F_(1,56)_ = 43.59, *p* < 0.0001). Saline injected E-sired males spent a significantly larger amount of time vertical in the OF center than saline injected C-sired males (*p* < 0.0001) and EtOH injected E-sired males (*p* < 0.0001). Similar results were observed for percentage of time vertical in the OF center (Figure 7D) with significant effects of sire (F_(1,56)_ = 31.21, *p* < 0.0001), treatment (F_(1,56)_ = 27.06, *p* < 0.0001), and interaction (F_(1,56)_ = 27.82, *p* < 0.0001). Pre-planned comparisons indicated that saline injected E-sired males spent a greater percentage of time vertical in the OF center compared to saline injected C-sired males (*p* < 0.0001) and EtOH injected E-sired males (*p* < 0.0001).

For female offspring on total vertical time in the OF (Figure 8A), a significant effect of treatment (F_(1,56)_ = 7.618, *p* < 0.01), but not sire or sire X treatment interaction was observed. However, secondary analysis found no significant differences between groups. Similarly, the average time spent vertical in the OF periphery (Figure 8B) was significantly affected by treatment only (F_(1,56)_ = 7.547, *p* < 0.01), but no differences in secondary analysis were observed. Time vertical in the center of the OF (Figure 8C) and percentage of time vertical in the OF center (Figure 8D) were not affected by treatment, sire, or the interaction of treatment X sire.

#### 3.3.2. Elevated Plus Maze

For male offspring on the EPM, there was no effect of sire, treatment, or sire X treatment interaction on total arm entries (Figure 9A), entries into open arms (Figure 9B), percentage of entries into open arms (Figure 9C), or percentage of time in open arms (Figure 9D).

For female offspring on the EPM, there was a significant effect of treatment on total arm entries (Figure 10A; F_(1,51)_ = 37.81, *p* < 0.01), entries into the open arms (Figure 10B; F_(1,51)_ = 52.82, *p* < 0.0001), percentage of total entries into the open arms (Figure 10C; F_(1,51)_ = 27.33, *p* < 0.0001) and the percentage of time in the open arms (Figure 10D; F_(1,51)_ = 17.41, *p* < 0.0001). There were no sire or sire X treatment interactions observed for any measured variable. Pre-planned comparisons revealed that in C-sired female offspring, EtOH injection increased total arm entries (*p* < 0.0001), entries into open arms (*p* < 0.0001), percentage of open arm entries (*p* < 0.001), and percentage of time in open arms (*p* < 0.01). Similarly, in E-sired female offspring, EtOH injection increased total arm entries (*p* < 0.01), entries into open arms (*p* < 0.001), and percentage of open arm entries (*p* < 0.01). 

### 3.4. Drinking in the Dark Assay

The DID assay was used to measure binge-like drinking behavior. EtOH consumption by male offspring on the training days (2 h sessions; days 1–3) and on the test day (4 h session; day 4) are shown in Figure 11A. E-sired male mice had significantly reduced EtOH consumption on the test day compared to C-sired males (Figure 11B; *p* < 0.05). BECs of samples collected at the end of the 4 h test day drinking session indicated that E-sired males also had lower BECs compared to C-sired males (Figure 11C; *p* < 0.05).

EtOH consumption by female offspring across the 4 days is shown in Figure 12A. There were no significant differences between C-sired females and E-sired females for EtOH consumption on the test day (Figure 12B) or BECs (Figure 12C). 

### 3.5. 2BC Drinking Assay

EtOH consumption and preference were measured across multiple EtOH concentrations in a continuous, home cage, unlimited access 2BC drinking paradigm. For male offspring, repeated measures ANOVA revealed an effect of both sire (F_(13,364)_ = 4.268; *p* < 0.05) and concentration (F_(13,364)_ = 3.112; *p* < 0.001) on total volume consumed (Figure 13A), but no effect of sire X concentration interaction. EtOH consumed (Figure 13B) indicated an effect of concentration (F_(13,364)_ = 26.3; *p* < 0.001), but no effect of sire or concentration X sire interaction. EtOH preference (Figure 13C) was affected by concentration (F_(13,364)_ = 6.339; *p* < 0.0001), but unaffected by sire or sire X concentration interaction. Body weight was significantly affected by time (Figure 13D; F_(6,168)_ = 17.53; *p* < 0.0001), but not by sire or sire X time interaction. Saccharin (Figure 13E) and quinine (Figure 13F) preference did not differ between C-sired and E-sired male offspring. 

For female offspring, repeated measures ANOVA revealed an effect of concentration on EtOH consumption (Figure 14B; F_(13,364)_ = 33.86; *p* < 0.0001), but no effect of sire or concentration by sire interaction. Total volume consumed (Figure 14A) was unaffected by sire, concentration, or sire X concentration interaction. EtOH preference (Figure 14C) did show a significant effect of concentration (F_(13, 364)_ = 10.23; *p* < 0.0001), but no effect of sire, or sire X concentration interaction. Body weight was significantly affected by time (Figure 14D; F_(6,168)_ = 45.66; *p* < 0.0001), but not by sire or sire X time interaction. Saccharin (Figure 14E) and quinine (Figure 14F) preference did not differ between C-sired and E-sired female offspring.

## 4. Discussion

In this study we tested the hypothesis that paternal preconception EtOH exposure via voluntary oral intake would have effects that persist across generations and impact anxiety-like and motor behavior, EtOH drinking, and behavioral sensitivity to EtOH in offspring. In support of this hypothesis, we observed effects of paternal preconception EtOH on several basal and EtOH-induced behaviors that were quantified in the OF behavioral assay. These effects were dependent on sex and behavioral endpoint. We also observed that paternal preconception EtOH exposure reduced EtOH binge-like drinking in male, but not female offspring. No effects of paternal preconception EtOH exposure were observed on the EPM assay, continuous access 2BC EtOH drinking assay, or for preference for the sweet and bitter tastants, saccharin and quinine, respectively.

Previous work from our laboratory has utilized a CIE vapor paternal exposure model [13,30,31,32,37]. These studies have consistently demonstrated that paternal preconception EtOH exposure impacts male, but not female F1 offspring. E-sire males had reduced EtOH preference and consumption in a 2BC drinking assay [30,31] and enhanced anxiolytic response to EtOH [30,31]. We have also observed that this exposure model alters stress responses in male offspring [37] that are similar to those observed by others following paternal preconception stress exposure [22,38]. CIE vapor exposure is stressful and activates the HPA axis (reviewed in [39]). We subsequently demonstrated that paternal preconception stress induced alterations in EtOH drinking and behavioral sensitivity to EtOH in offspring that were similar to that observed following paternal preconception EtOH exposure [32].

The model utilized here, voluntary oral self-administration is thought to be less stressful to the mice than the forced vapor exposure model used in previous studies. Many, but not all, results from the current study are consistent with our prior studies of paternal vapor exposure and paternal stress. All studies demonstrated an effect of paternal EtOH or stress on EtOH drinking in only the males of the next generation. EtOH drinking was selectively reduced in male offspring following paternal vapor EtOH [30,31], EOD EtOH (current study Figure 11), or stress [32]. Female drinking was unchanged by any of these paternal manipulations. The male specific changes in EtOH drinking were not due to a change in response to sweet or bitter taste sensitivity as preference for saccharin and quinine were not impacted by paternal EtOH or stress exposures (Figure 13E,F; [30,31,32]). Thus, irrespective of the paternal manipulation, E-sired and stress-sired male offspring seem to be protected from excessive EtOH drinking compared to C-sired offspring. Also consistent with previous studies [30,31], we did not observe an effect of paternal EtOH exposure on total locomotor activity following saline or EtOH injection in either sex in the OF assay (Figure 3A and Figure 4A).

However, not all results from the current study are consistent with results from our previous experiments. While paternal preconception EOD EtOH drinking changed DID drinking behavior in male offspring, it did not change EtOH preference or consumption of male offspring in a home cage, unlimited access 2BC drinking assay (Figure 13B,C). These results are inconsistent with our previous studies where forced paternal vapor exposure led to reduced EtOH consumption and preference in E-sired male offspring in an unlimited access 2BC drinking assay [30,31]. We did not test for DID EtOH consumption in our prior studies. It’s also important to note that the mice tested for DID in the current study were previously exposed to EtOH during the OF and EPM assays whereas those tested for unlimited access 2BC drinking were exposed to saline. This methodological difference could potentially explain the different outcomes of the DID and 2BC assays reported here. In the current study, we also did not observe an effect of paternal preconception EOD EtOH drinking on anxiolytic responses to EtOH as measured on the EPM. Previously, following paternal vapor exposure we consistently observed an enhanced anxiolytic response to EtOH that was male specific [30,31]. Thus, it appears that only a subset of behavioral alterations that are induced by forced paternal vapor exposure are also observed following voluntary EOD self-administration.

A potential explanation for the different outcomes is the impact of stress on the sires. Forced EtOH vapor exposure is thought to be stressful to the animals. In addition to direct effects of EtOH on a variety of stress responsive pathways (reviewed by [39]), vapor exposure entails the movement of animals in and out of the vapor chambers twice a day during each exposure day. In addition, each animal must be captured and briefly restrained so that a priming injection of EtOH can be administered at the start of each exposure session. Collectively, this is likely to be very stressful to the animals. In contrast, for voluntary EOD drinking, animals remain in their home cage, are never restrained, and drinking sessions are initiated simply by swapping a water drinking tube for one that contains EtOH. Although we have no data that directly address this issue, intuitively we speculate that the more stressful forced vapor exposure method induces a more robust phenotype in the next generation than the less stressful voluntary consumption model.

A second potential explanation for the differences between the current study and our previous results is that the different EtOH exposure paradigms resulted in different levels of EtOH exposure. The forced vapor exposure method results in BECs of ~150 mg/dL that are consistently sustained for 8 h/day, 5 days/week, for 5 weeks. In contrast, with the EOD drinking model used here, animals have voluntary access to EtOH for three 24 h periods every week for 12 weeks. Although we did not measure BECs in these animals because we wanted to minimize stress to the animals and because BECs are likely to vary in relation to the timing of EtOH consumption, others have reported maximum BECs of only 100 mg/dL using this method [34]. Thus, the two exposure methods result in very different patterns, durations, and exposure levels with the forced vapor exposure method likely achieving substantially greater exposure to EtOH. This too could explain the more robust phenotype that was observed following forced vapor exposure vs voluntary consumption. 

Lastly, a major difference between studies involves the timing of EtOH exposure. Sires in the current study were exposed to EOD EtOH drinking as adolescents (from 4–16 weeks of age; Figure 1). In contrast, in our prior studies with vapor exposure, sires were exposed only in adulthood (8–13 weeks of age; [30,31,37,40]). There are well documented differences in EtOH sensitivity [33,41,42] and stress responses [43,44,45] between adolescents and adults. It is unclear if this difference in experimental design impacted outcomes.

As mentioned above, prior work [30,31] and that reported here did not find an effect of paternal EtOH exposure on total locomotor activity in the OF assay (Figure 3A and Figure 4A). The current work extends our previous observations by examining several additional behavioral measures that we did not study previously. We found numerous, complex, sex-specific alterations in offspring that resulted from paternal EtOH consumption.

For male offspring following injection with saline, no differences were observed between C-sired and E-sired animals on measures of total locomotor activity (Figure 3A). However, when activity in the center of the maze was examined, we observed that E-sired animals spent significantly more time in the center area compared to C-sired offspring (Figure 3F). We also observed that E-sired males also spent significantly more time resting (Figure 5C) and rearing (Figure 7C) in the center portion of the OF apparatus. All these observations are consistent with a reduced basal level of anxiety-like behavior in the E-sired males compared to controls.

For male offspring following injection with 0.75 g/kg EtOH, we surprisingly observed that the acute EtOH treatment caused E-sired males to spend less time (Figure 3F), to rest less (Figure 5C), and to rear less (Figure 7C) in the center of the OF apparatus. This is surprising because low dose EtOH is typically anxiolytic, but these animals appear to have an anxiogenic response to EtOH. In contrast, C-sired animals were not sensitive to the acute EtOH treatment on activity (Figure 3D), resting (Figure 5C), or rearing (Figure 7C) in the center. Thus, paternal preconception EtOH exposure increased sensitivity to an acute injection of EtOH and paradoxically induced an anxiogenic-like response in male offspring.

These results of OF testing for anxiety-like behaviors are inconsistent with our EPM data in which no alterations in anxiety-like behavior were observed (Figure 9). While the EPM is the gold standard for measuring anxiety-like behavior in rodents, there is considerable disagreement as to the interpretation of OF behaviors and how they relate to emotionality [46] (and as reviewed in [47,48,49]). Thus, it is likely that the OF is measuring a different aspect of emotionality than the EPM. It is also possible that the timing of OF and EPM testing contributed to the inconsistent results. OF testing was started 10 min after EtOH injection whereas EPM was started 15 min after EtOH injection when effects of EtOH may have waned.

For female offspring in the OF assay, following saline injection there were no differences between C- and E-sired animals on any of the parameters analyzed (Figure 4, Figure 6 and Figure 8). Acute EtOH injection caused a small, but significant increase in total ambulatory activity (Figure 4A), activity in the periphery (Figure 4B), and a slight decrease in total resting time (Figure 6A) in the C-sired animals. E-sired females were unresponsive to the acute EtOH injection. Thus, paternal EOD drinking may reduce sensitivity to acute EtOH in females in the OF assay. However, the observed effects are very small and additional studies using optimized doses and timing of injection are needed to fully understand the impact of paternal EtOH on female offspring on this assay.

For female offspring in the EPM assay, no significant differences between C- and E-sired animals were found on any of the parameters analyzed (Figure 10). Acute EtOH injection was shown to increase all parameters for C-sired female mice and all but percentage of time spent in open arms for E-sired female mice. Thus, female offspring appear to have normal basal anxiety-like behavior and a normal anxiolytic response to acute EtOH injection.

In conclusion, the current study contributes to an expanding literature on the intergenerational effects of paternal EtOH. While numerous studies have been published that document an effect of paternal preconception EtOH exposure on a wide variety of developmentally related outcomes including basal behavioral responses (for reviews, see [5,25]), only recently have EtOH drinking and behavioral sensitivity to EtOH been investigated. The current study demonstrates that voluntary EOD EtOH drinking by male mice prior to mating can reduce binge-like EtOH consumption and increase behavioral sensitivity to EtOH on some endpoints selectively in male offspring. This adds to studies demonstrating that paternal preconception forced vapor exposure [30,31] also impact the EtOH phenotype of the next generation. Together these studies make a compelling argument that an individual’s behavioral response to EtOH and EtOH drinking behaviors are impacted by EtOH exposure of the prior generation. Understanding the putative epigenetic mechanisms behind this non-Mendelian mode of inheritance may provide novel insights that may illuminate points of intervention that could be exploited to combat AUD.

## Figures and Tables

**Figure 1 brainsci-09-00056-f001:**
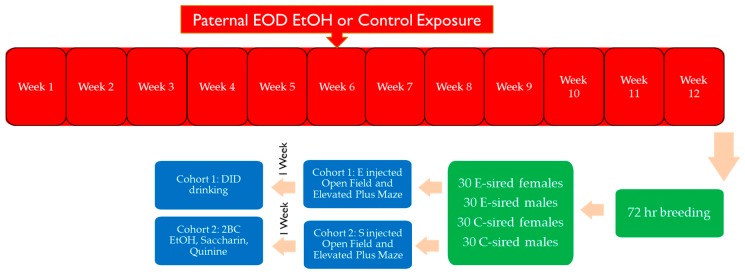
Experimental overview. F0 sections are marked in red, green denotes breeding, and blue denotes F1 generation behavioral assays.

**Figure 2 brainsci-09-00056-f002:**
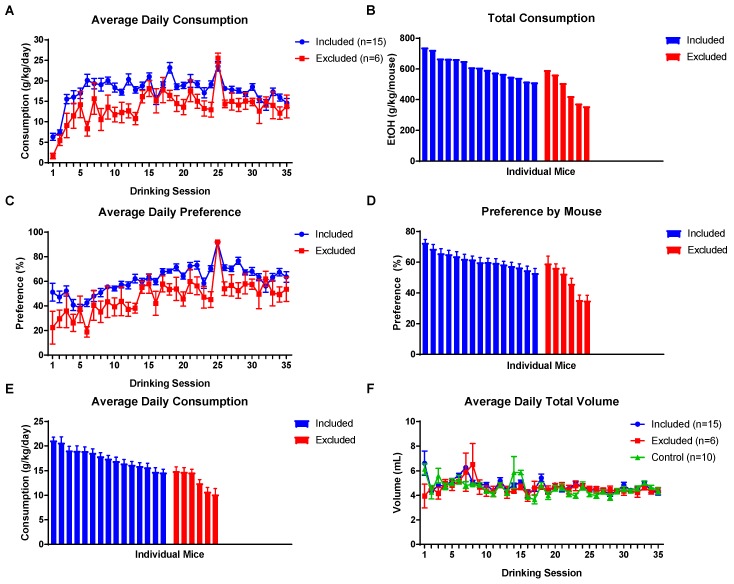
EOD 2BC drinking results of sires. (**A**) Average EtOH consumption per drinking session of those animals that were selected for breeding (included) and those that were not used for breeding (excluded). (**B**) Cumulative EtOH consumption for the ~3 months exposure period for individual mice in the included and excluded groups. (**C**) Average EtOH preference per session by group. (**D**) EtOH preference of individual mice in the included and excluded groups. (**E**) Individual average daily EtOH consumption for the ~3 months exposure period observed for individual mice in the included and excluded groups. (**F**) The average total volume of fluid ingested per session by mice did not differ between groups. Data presented as mean ± SEM. (Some error bars in (**A**,**C**,**F**) are obscured by data points.).

**Figure 3 brainsci-09-00056-f003:**
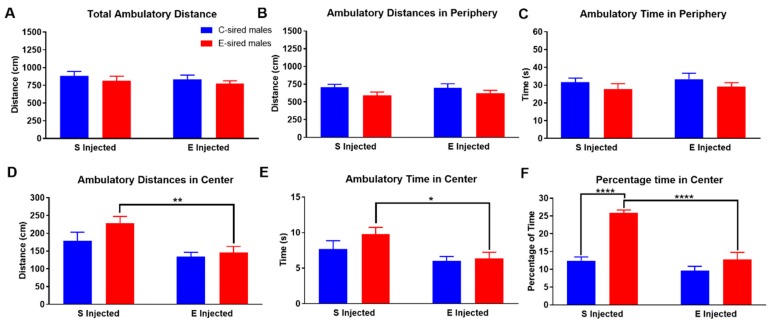
Male OF ambulatory results. Total ambulatory distance (**A**), ambulatory distance in the periphery (**B**), and ambulatory time in periphery (**C**) were similar in all groups. (**D**) EtOH injected E-sired males had reduced ambulatory distance in the center of the OF compared to saline injected E-sired males. (**E**) Saline injected E-sired males spent significantly more ambulatory time in the center of the OF cage compared to EtOH injected E-sired males. (**F**) Saline injected E-sired males spent significantly greater amount of time in the center of the OF cage compared to saline injected C-sired males and EtOH injected E-sired males. Data presented as mean ± SEM; * *p* < 0.05, ** *p* < 0.01, **** *p* < 0.0001; *n* = 15 per group.

**Figure 4 brainsci-09-00056-f004:**
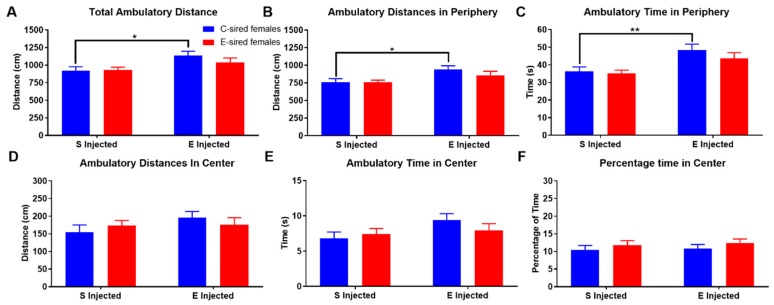
Female OF ambulatory results. EtOH injected C-sired females had greater total ambulatory distance (**A**) and ambulatory distance in the periphery (**B**) compared to saline injected C-sired females. (**C**) EtOH injected C-sired females had greater ambulatory time in periphery compared to saline injected C-sired females. Ambulatory distance in the center (**D**), ambulatory time in center (E), and percent time in center (**F**) were similar in all groups. Data presented as mean ± SEM; * *p* < 0.05, ** *p* < 0.01; *n* = 15 per group.

**Figure 5 brainsci-09-00056-f005:**
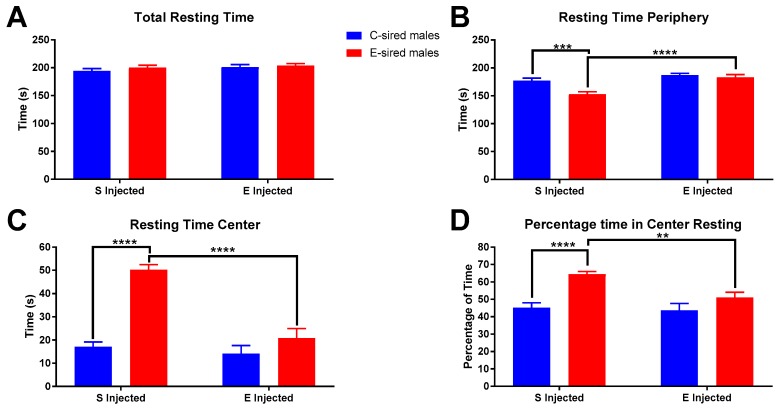
Male OF resting results. (**A**) Total resting time was not significantly different between groups. (**B**) Saline injected E-sired males spent less time resting in the periphery compared to saline injected C-sired males and EtOH injected E-sired males. (**C**) Saline injected E-sired males spent more time resting in the OF center compared to both saline injected C-sired males and EtOH injected E-sired males. (**D**) Saline injected E-sired males spent significantly more time resting in the center compared to both saline injected C-sired males and EtOH injected E-sired males. Data presented as mean ± SEM; ** *p* < 0.01, *** *p* < 0.001, **** *p* < 0.0001; *n* = 15 per group.

**Figure 6 brainsci-09-00056-f006:**
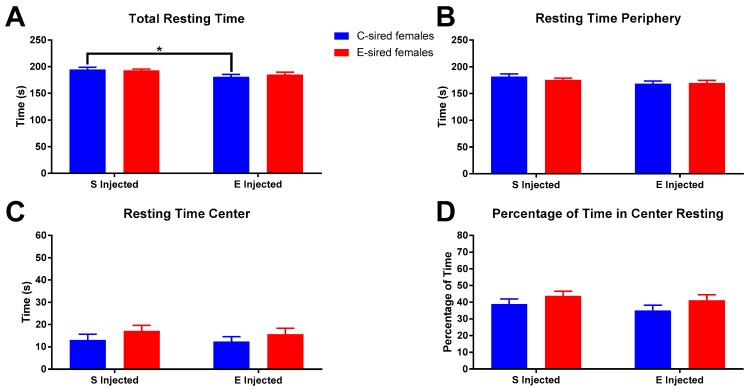
Female OF Resting results. (**A**) Saline injected C-sired females had significantly greater total resting time compared to EtOH injected C-sired females. (**B**) There was an overall effect of EtOH injection on resting time in the periphery, but no significant differences in pre-planned comparisons were observed. Time resting in center (**C**) and percentage of time resting in center (**D**) were similar in all groups. Data presented as mean ± SEM; * *p* < 0.05; *n* = 15 per group.

**Figure 7 brainsci-09-00056-f007:**
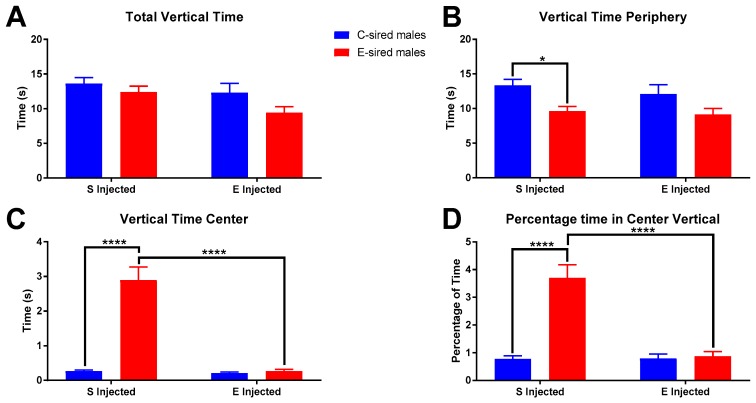
Male OF vertical results. (**A**) Total vertical time was significantly affected by sire and treatment, but not interaction of sire X treatment and secondary analysis did not reveal any group differences. (**B**) Saline injected E-sired males spent significantly less time vertical in the periphery compared to saline injected C-sired males. (**C**) Saline injected E-sired males spent significantly more time vertical in the center compared to both saline injected C-sired males and EtOH injected E-sired males. (**D**) Saline injected E-sired males spent a significantly greater percentage of time vertical in the center compared to both saline injected C-sired males and EtOH injected E-sired males. Data presented as mean ± SEM; * *p* < 0.05, **** *p* < 0.0001; *n* = 15 per group.

**Figure 8 brainsci-09-00056-f008:**
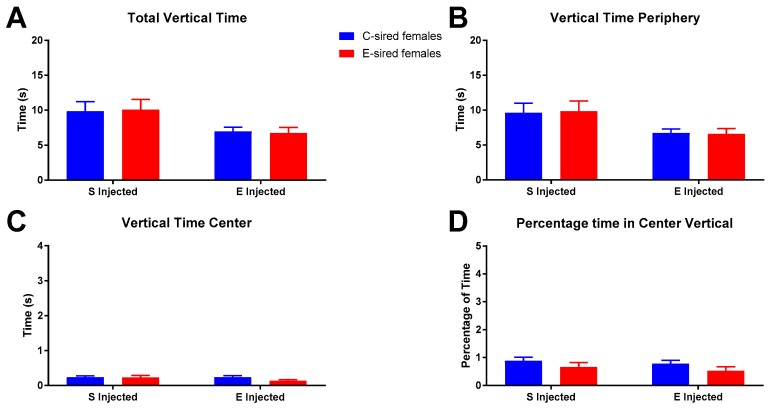
Female OF vertical results. Significant overall effects of treatment on total vertical time (**A**) and vertical time in periphery (**B**) were observed, but no differences in secondary analysis were detected. Vertical time in center (**C**) and percentage time vertical in center (**D**) were similar between all groups. Data presented as mean ± SEM; *n* = 15 per group.

**Figure 9 brainsci-09-00056-f009:**
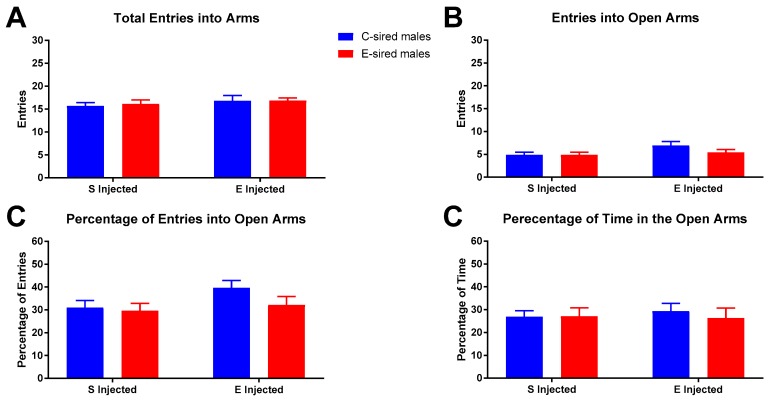
Male EPM results. No significant differences between groups were observed for total arm entries (**A**), entries into open arms (**B**), percentage open arm entries (**C**), or percentage of time in open arms (**D**). Data presented as mean ± SEM; *n* = 13, saline injected C-sired and saline injected E-sired; *n* = 15, EtOH injected C-sired; *n* = 14, EtOH injected E-sired.

**Figure 10 brainsci-09-00056-f010:**
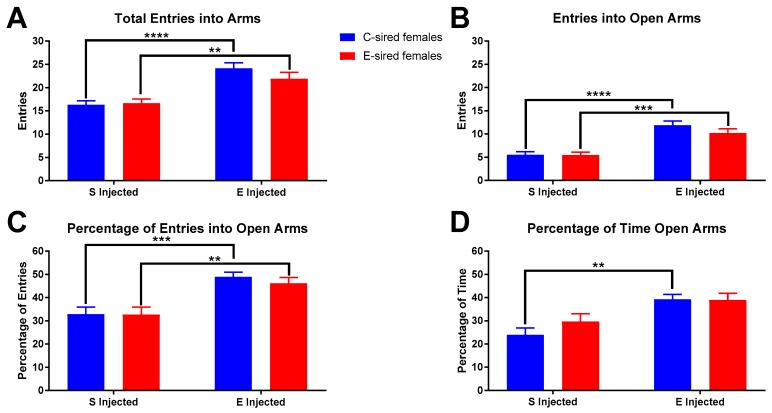
Female EPM results. Compared to saline treatment, EtOH injection increased total arm entries (**A**), entries into open arms (**B**), percentage open arm entries (**C**), and percentage of time in open arms (**D**) in C-sired and E-sired groups. Data presented as mean ± SEM; ** *p* < 0.01, *** *p* < 0.001, **** *p* < 0.0001; *n* = 15, saline injected C-sired and saline injected E-sired; *n* = 13, EtOH injected C-sired and EtOH injected E-sired.

**Figure 11 brainsci-09-00056-f011:**
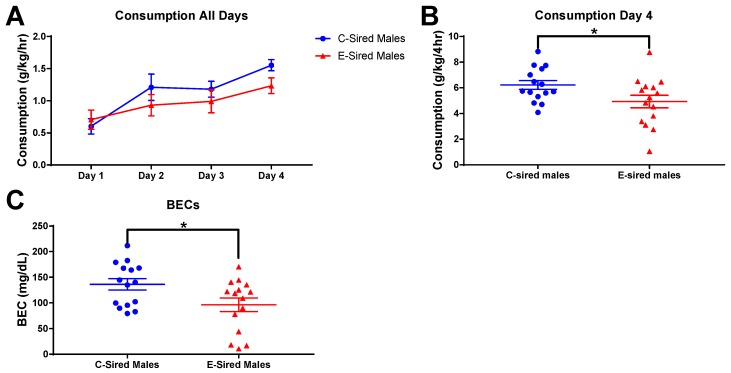
Male DID results. (**A**) EtOH consumption on the three two hour training sessions (Days 1–3) and the four hour test session (Day 4). (**B**) Day 4 EtOH consumption was reduced in E-sired males compared to C-sired males. (**C**) BECs of E-sired males were significantly lower compared to C-sired males. Data presented as mean ± SEM; * *p* < 0.05; *n* = 15 per group.

**Figure 12 brainsci-09-00056-f012:**
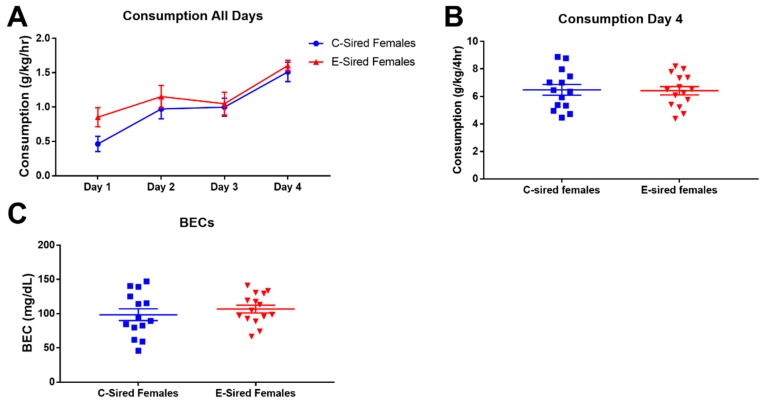
Female DID results. (**A**) EtOH consumption on the three two hour training sessions (Days 1–3) and the four hour test session (Day 4). There was no significant difference on Day 4 in EtOH consumption (**B**) or BECs (**C**) between C-sired and E-sired females. Data presented as mean ± SEM; *n* = 14 C-sired; *n* = 15, E-sired.

**Figure 13 brainsci-09-00056-f013:**
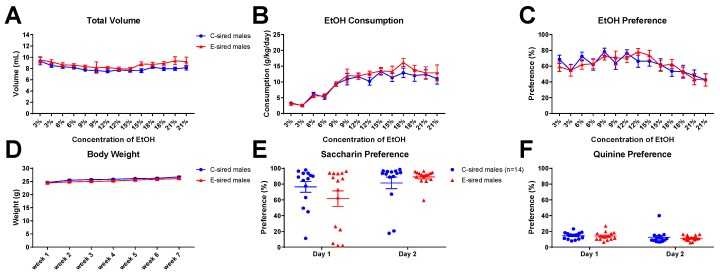
Male 2BC results. (**A**) The total volume of fluid (water plus EtOH) consumed was affected by both sire (*p* < 0.05) and concentration (*p* < 0.001). (**B**) EtOH consumption was affected by concentration (*p* < 0.001) but not by sire or interaction. (**C**) Preference was affected by concentration (0.0001) but not sire, or interaction. Body weight (**D**), saccharin preference (**E**), and quinine preference (**F**) were not affected by sire. Data presented as mean ± SEM; *n* = 15 per group.

**Figure 14 brainsci-09-00056-f014:**
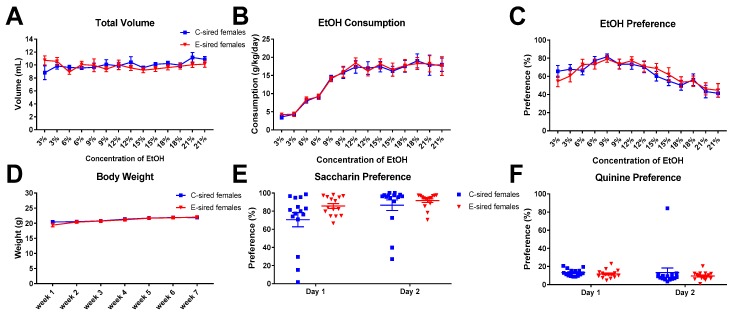
Female 2BC results. (**A**) The total volume of fluid (water plus EtOH) consumed was not affected by sire, concentration or interaction. EtOH consumption (**B**) and preference (**C**) were affected by concentration (*p* < 0.0001 and *p* < 0.0001, respectively) but not by sire or interaction. Body weight (**D**), saccharin preference (**E**), and quinine preference (**F**) were not affected by sire. Data presented as mean ± SEM; *n* = 15 per group.

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
