# Peer review of "Paternal Preconception Every-Other-Day Ethanol Drinking Alters Behavior and Ethanol Consumption in Offspring"

_brainsci, 2019, doi:10.3390/brainsci9030056_

Round 1
Reviewer 1 Report
In this manuscript, Beeler and colleagues examined the impact of preconception sire ethanol drinking on anxiety-like behaviors, EtOH sensitivity and EtOH consumption in male and female offspring. The clearest and main finding of this comprehensive series of experiments is that EtOH sire drinking reduces EtOH binge-like drinking in male but not female offspring. Sire EtOH drinking also had complex, sex-specific effects on anxiety-like behavioral assays in the next generation. The experiments were properly designed, the results are novel and interesting and the manuscript is clear and well organized. These findings confirm and extend previous reports from this research group regarding the multigenerational impact of ethanol exposure in sires on the next generation of animals. One novel component in this study was the volitional consumption of EtOH by the sires (as opposed to experimenter-administered/vapor EtOH exposure). I mostly have minor comments and questions, outlined below.
Did EtOH exposure impact litter size or sex ratio?
In the statistical analysis section, report the within-subject and between subject factors for the repeated measures ANOVAs.
First paragraph of results section. I suggest the author refer to Figure 1 first in the first sentence. The authors describe selecting high EtOH drinking sires for breeding using Figure 3, it should be Figure 2 (i.e. line 181).
Question mark instead of +/- in the legend of Figure 2
Figure 11, it would be helpful to make the asterisks larger and move them away from the lines going across the plots.
line 321, what is meant by "basic behavior"? Suggestions include "anxiety-like and motor behaviors.
Line 364,I suggest replacing "obvious" with "potential". Same for line 375.
Author Response
We thank the reviewers for their thoughtful suggestions and have the revised the manuscript accordingly. Our responses are shown in italics below each comment.
Referee: 1
Comments to the Author
I mostly have minor comments and questions, outlined below.
1. Did EtOH exposure impact litter size or sex ratio?
They did not. We have included this information within section 3.2 (lines 194-195).
2. In the statistical analysis section, report the within-subject and between subject factors for the repeated measures ANOVAs.
We have included that the within-subject factors for the repeated measures ANOVAs for body weight and 2BC analysis were time and concentration, respectively (lines 174-175).
3. First paragraph of results section. I suggest the author refer to Figure 1 first in the first sentence. The authors describe selecting high EtOH drinking sires for breeding using Figure 3, it should be Figure 2.
A reference to Figure 1 has been included to further explain the sire timeline (line181). The mistaken labeling has been fixed, using Figure 2 rather than Figure 3 in order to describe sire selection (lines 182-186).
4. Question mark instead of +/- in the legend of Figure 2
We replaced the question mark with +/-.
5. Figure 11, it would be helpful to make the asterisks larger and move them away from the lines going across the plots.
To better enable visibility of the asterisks, we have both increased their size and moved them further away from the demarcating lines, as requested.
6. line 321, what is meant by "basic behavior"? Suggestions include "anxiety-like and motor behaviors.
Basic behaviors include those anxiety-like and motor behaviors. In order to better communicate our findings, we have replaced “basic behavior” with the suggested specifications (lines 3236-324).
7. Line 364, I suggest replacing "obvious" with "potential". Same for line 375.
Replaced (lines 369, 380).
Reviewer 2 Report
The article titled “Paternal preconception every-other-day ethanol drinking alters behavior and ethanol consumption in offspring” by Beeler et al., (2019) investigated the effects of voluntary ethanol drinking by rodent fathers on offspring behavior and alcohol drinking in the dark. The work is incredibly interesting and well thought out. The article is clear in its methodology and in its conclusions. Furthermore, in this reviewer’s opinion voluntary drinking is much more representative of what is happening in humans and is therefore a wonderful model. I applaud the authors for this lengthy and fund intensive ethanol exposure model. This reviewer does have just a few questions.
1) What was criteria for excluding sires? I see some excluded drank a good bit so it couldn’t have been only because of low drinking. Was it that they simply did not yield offspring?
2) In section 3.1 I think that Fig. 2 rather than 3 should be referenced.
3) One methodological issue I see is that the groups were split into 2 cohorts and cohort 1 had ethanol injections during OF and EPM testing while cohort 2 had saline. What this means is that the DID animals had prior ethanol exposure. Would it not have been better to use half the animals from Cohort 1 and 2 in this and then the other half in the choice test? In either case this should be mentioned in the discussion as this previous exposure could have affected DID.
4) Concerning the vertical time in the OF, so is it the injection alone having an effect in results for vertical time in males? It seems that it is always the E-sired Saline animal that shows an effect. Perhaps the ethanol is counteracting the stress of the injection and leaving these animals unchanged from controls? I am not suggestion that this needs to be mentioned but perhaps it should be considered.
In general I found this manuscript incredibly well written.
Author Response
We thank the reviewers for their thoughtful suggestions and have the revised the manuscript accordingly. Our responses are shown in italics below each comment.
Referee: 2
Comments to the Author
This reviewer does have just a few questions.
1. What was criteria for excluding sires? I see some excluded drank a good bit so it couldn’t have been only because of low drinking. Was it that they simply did not yield offspring?
Those animals that were excluded either had consumption/preference that were on the lower end, or they had consumption/preference that was highly variable from day to day. Thus, the animals that were selected for breeding had consistently high levels of consumption/preference. This is stated in the methods section (lines 115-116).
2. In section 3.1 I think that Fig. 2 rather than 3 should be referenced.
You are correct and the noted references have been changed.
3. One methodological issue I see is that the groups were split into 2 cohorts and cohort 1 had ethanol injections during OF and EPM testing while cohort 2 had saline. What this means is that the DID animals had prior ethanol exposure. Would it not have been better to use half the animals from Cohort 1 and 2 in this and then the other half in the choice test? In either case this should be mentioned in the discussion as this previous exposure could have affected DID.
Our rationale for this approach was two fold. First, we felt that a single low dose injection of ethanol 1 week prior to DID would have minimal impact on DID. Second, we believe that having all animals treated the same prior to DID (i.e., all animals received the ethanol injection) would reduce variability in the data as opposed to the alternative of having half the animals previously treated with ethanol and half with saline. We now mention this methodological difference in the discussion (lines 360-363).
4. Concerning the vertical time in the OF, so is it the injection alone having an effect in results for vertical time in males? It seems that it is always the Esired Saline animal that shows an effect. Perhaps the ethanol is counteracting the stress of the injection and leaving these animals unchanged from controls? I am not suggestion that this needs to be mentioned but perhaps it should be considered.
We have considered this possibility but without uninjected control-sired and ethanol-sired groups for comparison, we are reluctant to speculate on this possibility. Therefore, we have not made any changes to the manuscript in response to this comment.